# Cloning and Functional Analysis of *NoMYB60* Gene Involved in Flavonoid Biosynthesis in Watercress (*Nasturtium officinale* R. Br.)

**DOI:** 10.3390/genes13112109

**Published:** 2022-11-14

**Authors:** Xiaoqing Ma, Jiajun Ran, Guihu Mei, Xilin Hou, Xiong You

**Affiliations:** 1State Key Laboratory of Crop Genetics & Germplasm Enhancement, Key Laboratory of Biology and Genetic Improvement of Horticultural Crops (East China), Ministry of Agriculture and Rural Affairs of China, Engineering Research Center of Germplasm Enhancement and Utilization of Horticultural Crops, Ministry of Education of China, Nanjing Agricultural University, Nanjing 210095, China; 2Nanjing Suman Plasma Engineering Research Institute, Nanjing Agricultural University, Nanjing 210095, China; 3College of Sciences, Nanjing Agricultural University, Nanjing 210095, China

**Keywords:** MYB60, flavonoid synthesis, *Nasturtium officinale* R. Br., *NoMYB60*, ABA, SA, MeJA, transcriptional activation activity

## Abstract

The *MYB60* gene belongs to the R2R3-MYB subfamily, which includes the *MYB31/30/96/94* genes. Although these genes have been shown to respond to heat and drought stresses, their role in flavonoid synthesis remains unclear. In this study, *NoMYB60* was cloned from watercress and its structure and function were analyzed. Sequence structure analysis showed that *NoMYB60* had a highly conserved R2R3 DNA-binding region at the N-terminus. Under the treatment of ABA, SA or MeJA, the expression level of *NoMYB60* first significantly increased and then decreased, indicating that ABA, SA and MeJA positively regulated *NoMYB60*. The subcellular localization of NoMYB60-GFP indicated that NoMYB60 was localized in the nuclear region, which is consistent with the molecular characterization of the transcription factor. Gene silencing experiments were also performed to further test the function of *NoMYB60*. The result showed that virus-induced silencing of *NoMYB60* affected the expression of enzyme genes in flavonoid synthesis pathways and promoted the synthesis of flavonoids. Moreover, we discovered that NoMYB60 interacts with NoBEH1/2. In this study, provides a reference for research on the regulation mechanism of flavonoid synthesis in Cruciferae and other crops.

## 1. Introduction

Watercress (*Nasturtium officinale* R. BR.) is a cruciferous plant that is cultivated in many regions of China and also distributed in Asia, Europe and North America. The edible parts are the tender stems and leaves of the plant [1]. Flavonoids, a kind of polyphenols, which are found in almost all leaves, flowers, fruits and roots, are important for plant growth and stress resistance. Flavonoids are a vast class of secondary metabolites encompassing more than 10,000 structures [2], many of which are present in high amounts in vegetables and fruits. Structural genes and regulatory genes encoding major enzymes in flavonoid biosynthesis pathways have been identified in many crops [3,4]. The structural genes include phenylalanine ammonia-lyase (PAL), 4-coumarate:CoA ligase (4CL), cinnamate 4-hydroxylase (C4H), chalcone synthase (CHS), flavanone 3-hydroxylase (F3H), chalcone isomerase (CHI), flavonoid 3′-hydroxylase (F3′H), flavonol synthase (FLS), anthocyanin synthetase (ANS), dihydroflavonol 4-reductase (DFR), aUDP-glucose:flavonoid 3-O-glucosyltransferase (UFGT), etc. [5].

MYB-class transcription factors, referred to a class of transcription factors containing the MYB domain, play a key role in regulating flavonoid biosynthesis and stress tolerance. As a member of the R2R3-MYB family in *Arabidopsis thaliana, MYB60* is characterized by a reduced defense-cell-specific expression pattern under drought stress [6]. In addition, overexpression of *MYB44*, another protective-cell-specific member of the MYB family, is associated with enhanced ABA sensitivity and faster stomatal closure compared with wild-type plants, resulting in reduced expression of the gene-encoding type 2C protein phosphatase (a known negative regulator) [7]. This partly explains the enhanced tolerance of *MYB44*-overexpressing plants to abiotic stress. *MYB61* is another MYB superfamily member, also belonging to the R2R3-MYB family, and is involved in the response leading to stomatal closure [8]. Seo et al. found that *MYB96-*mediated ABA signaling can be integrated into the auxin signaling pathway [9]. ABA induced the expression of *MYB96* but inhibits the expression of *MYB60*. These results indicate that the expression of each gene in the MYB family is delicately controlled, with specificity.

*MYB60* genes can improve the tolerance of seedlings to high salt stress [10]. For example, overexpression of cotton *GbMYB60* in *Arabidopsis* increased salt sensitivity of transgenic *Arabidopsis* [11]. Lv Y et al. identified and expressed the MYB gene family of *Koelreuteria bipinnata Franch* and predicted that the *KbMYB60* gene negatively regulates anthocyanin synthesis [12]. In this study, the *MYB60* gene was identified from watercress, named *NoMYB60*, which has a highly conserved R2R3 domain and is expressed in the nucleus. Under exogenous ABA, SA or MeJA treatment, the transcription level of *NoMYB60* first gradually increased and then decreased. In order to further test the function of *NoMYB60*, gene silencing experiments were carried out. After *NoMYB60* gene silencing, the key genes of the flavonoid synthesis pathway were upregulated or downregulated to varying degrees. The interaction between NoMYB60 and NoBEH1/2 was confirmed by yeast two-hybrid and bimolecular fluorescence complementation techniques. These results indicated that *NoMYB60* is involved in the biosynthesis of flavonoids in watercress.

## 2. Materials and Methods

### 2.1. Plant Material

Plants of the watercress cultivar used in this study were provided by the Chinese Cabbage System Biology Laboratory of Nanjing Agricultural University. The material was imported from the US, where it had been bred by Seed Needs Company “https://www.myseedneeds.com/ (accessed on 3 November 2018)”. All plants were grown in a climatic chamber at 24 °C/18 °C with a 16 h/8 h light/dark cycle.

Before the hormone treatment, watercress seedings of length about 15 cm were cut and cultivated in the Hoagland nutrient solution. The watercress root was immersed in approximately 2 cm of nutrient solution. After 15 days, the plants were divided into three groups: SA group, ABA group and MeJA group. Volumes of 5 mg/L SA, 5 mg/L ABA and 1 mg/L MeJA were used for hydroponic treatment, respectively. Leaf samples were collected after 0 h, 2 h, 4 h, 6 h, 12 h, 24 h, 36 h and 48 h. All collected samples were placed in the refrigerator at −80 °C for use.

For the virus-induced gene silencing (VIGS) assay, watercress with a length of about 15 cm was cut and cultured in nutrient solution. After 15 days, NoMYB60-silenced plants were obtained using gene gun-mediated transformation (1300 psi, PDS-1000/He, Bio-Rad, Hercules, CA, USA).

### 2.2. Sequence Analysis

The PlantCARE website “http://bioinformatics.psb.ugent.be/webtools/plantcare/html/ (accessed on 5 August 2020)” was used to analyze the cis-acting elements of the promoter, and the result is shown in Table 1. The ExPASy website “https://web.expasy.org/protparam/ (accessed on 10 August 2020)” was used to analyze the physical and chemical properties of NoMYB60 protein; DNAMAN software to analyze the amino acid multi sequence alignment; and the MEME website to analyze the amino acid sequence conservative motif. The phylogenetic tree was constructed using MEGA 7.0 by the neighbor-joining method with 1000 bootstrap replicates.

### 2.3. Quantitative Real-Time PCR Analysis

The total RNA of the plant samples was isolated using the RNA Simple Total RNA Kit (TIANGEN, Beijing, China). The RNA purity and concentration were determined using a Micro-Spectrophotometer (ALLsheng, Hangzhou, China). The integrity of RNA was verified by performing 1.2% agarose gel electrophoresis. An amount of 1 μg of total RNA was used for cDNA synthesis with HiScript III RT SuperMix for qPCR (+gDNA wiper) (Vazyme, Nanjing, China). The full-length cDNA sequences of *PAL*, *C4H*, *4CL*, *CHs*, *F3H*, *DFR*, *ANS*, *UFGT* and *NoMYB60* were isolated by 5′ and 3′ RACE with PrimeSTAR^®^ Max DNA Polymerase (Takara, Beijing, China). RT-qPCR experiments were performed on an Applied Biosystems StepOnePlus^TM^ Real-Time PCR System Thermal Cycling Block using Hieff UNICON^®^ qPCR SYBR Green Master Mix (Yeasen, Shanghai, China). The reactions were performed in a 20 µL total volume containing 10 µL of Hieff UNICON^®^ qPCR SYBR Green Master Mix, 1.0 µL of diluted cDNA, 0.4 µL of each primer and 7.2 µL ddH_2_O. The following amplification program was used: 95 °C for 30 s, followed by 40 cycles of 95 °C for 10 s, 60 °C for 20 s and 72 °C for 20 s in 96-well quantitative PCR plate (Mulinsen, Nanjing, China). To analyze the melting curve for determine primer specificity, we added a procedure of 95 °C for 15 s and 60 °C for 1 min, followed by 95 °C for 15 s. Specific primers (Appendix A) for qRT-PCR analysis were designed with Primer3Plus “https://www.primer3plus.com/index.html (accessed on 25 August 2020)”.

### 2.4. Determination of Total Flavonoid Content

The sample was dried to a constant weight, pulverized, and passed through a 40-mesh sieve, weighed about 0.02 g. 2 mL of extract was added and shaken at 60 °C for 2 h and then centrifuged at 10,000 rpm, rest at 25 °C for 10 min. Finally, the supernatant was collected for further analysis (Suzhou Comin Biotechnology Co., Ltd., Suzhou, China).

### 2.5. Virus-Induced Gene Silencing (VIGS)-Mediated Silencing of NoMYB60 in Watercress

A specific 40 bp fragment was selected from the coding region of *NoMYB60*, combined with its antisense sequence to synthesize and insert into a PTY-S (PTY) vector. PTY is a carrier of the TYMV-VIGS system. The synthesis of fragments and the construction of silencing vectors were performed by the GeneScript company (Nanjing, China). The hairpin structural sequence of VIGS is listed in Appendix A. Watercress of length about 15 cm was used in this VIGS test. The 4 µg PTY and NoMYB60-PTY plasmids were wrapped in gold particles, and then four watercress plants were bombarded using the gene-gun-mediated transformation (1300 psi, PDS-1000/He, Bio-Rad, Hercules, CA, USA). Two weeks later, the leaves showing virus-mottled leaf surfaces were sampled for detection.

### 2.6. Transcriptional Activation Activity Assays

The transcriptional activation activity of NoMYB60 was studied using a yeast two-hybrid (Y2H) system. The full-length cDNA of *NoMYB60* was fused to the 3′ end of the GAL4 DNA binding domain of the PGBKT7(BD) vector to obtain the BD-NoMYB60 construct. The BD-NoMYB60 construct was co-transformed with the empty pGADT7 vector (AD) into the yeast strain Y2H gold.

Subsequently, according to the functional domain of the NoMYB60 protein sequence, the sequence was divided into two segments at the 227th amino acid and constructed into the BD vector. BD-NoMYB60 (full-length sequence) and BD-NoMYB60-1 (1–227, amino acids) were obtained and stored at −20 °C for later use. See Appendix A for specific primers for constructing the BD vector. Identify transcriptional activation domains (TADs) as described above.

### 2.7. Yeast Two-Hybrid (Y2H) Assays

The recombinant plasmid BD-NoMYB60-1 was co-transferco-transferred into yeast competent cells with AD-NoBEH1 and AD-NoBEH2, respectively, plotted on SD/-Trp/-Leu medium and cultured for 2–3 days. A single colony was selected and transferred to YPDA liquid medium for overnight culture. The cloning primers of the gene were used for PCR detection. The OD of bacterial solution was adjusted to 0.8–1.0, and 8 μL bacterial solution was cultured on SD/-Leu/-Trp and SD/-Ade/-His/-Leu/-Trp medium at 28 °C for 3–4 days. Then the cell growth was observed. The same operation was performed on the control.

### 2.8. Bimolecular Fluorescence Complementary (BiFC) Assays

The coding sequence of NoMYB60 was amplified with primers YC-NoMYB60-F and YC-NoMYB60-R (Appendix A) and cloned into the pUC-SPYCE vector to generate NoMYB60-cYFP plasmids. The coding sequence of NoBEH1/2 was amplified by primers YN-NoBEH1/2-F and YN-NoBEH1/2-R (Appendix A) and cloned into pUC-SPYNE to generate NoBEH1-nYFP and NoBEH2-nYFP. The plasmid mixtures (NoMYB60-cYFP and NoBEH1-nYFP, NoMYB60-cYFP and NoBEH1-nYFP, NoMYB60-cYFP, nYFP) were introduced into tobacco cells. After 24–48 h incubation, the cells were observed and photographed with a laser confocal microscope.

## 3. Results

### 3.1. Identification of NoMYB60 and Sequence Alignment

The cDNA sequence of *NoMYB60* was isolated from watercress, encoding a protein of 281 amino acids with a predicted molecular mass of 32.01 kDA and theoretical pI of 5.79. It is an unstable hydrophobic protein. Amino acid multiple sequence alignment of MYB60 proteins of different species, including *Arabidopsis* (At), pak choi (*Brassica campestris* (syn. *Brassica rapa*) ssp. *Chinensis*; Bc), *N. officinale* (No), *Oryza sativa* (Os), *Zea mays* (Zm), *Solanum lycopersicum* (Sl), *Daucus carota* (Dc) and *Triticum aestivum* (Ta), showed that the N-terminal of the MYB60 protein has a highly conserved R2R3 domain and two unique conserved regions, Box1 and Box2 (Figure 1A).

Furthermore, the results from multiple sequence alignment showed that the NoMYB60 protein shares 48.92%, 42.57%, 42.68% and 48.92% identity with OsMYB306-1, ZmMYB306, TaMYB60-like and DcMYB306, respectively, while it shares 86.62% and 80.43% identity with AtMYB60 in *Arabidopsis* and BcMYB60 in pak choi, which further confirmed the conservation of cruciferous plants in the evolutionary process. The NCBI accession numbers of genes are shown in Appendix A.

Motif distribution studies showed that motif 1, motif 2, motif 4 of N-terminal and motif 3 and motif 5 of C-terminal were highly conserved among all MYB60 proteins (Figure 1B). In addition, PSIPRED 4.0 was used to predict the secondary structure. It was found that the NoMYB60 protein is mainly composed of α-helix (98 amino acids, 35%) and irregular curl (182 amino acids, 65%).

### 3.2. Phylogenetic Analysis of MYB60

In order to analyze the phylogenetic relationship of MYB60 protein among species, an unrooted neighbor-joining (NJ) phylogenetic tree was constructed with MEGA 7.0 software, including *S. lycopersicum*, *O. sativa*, *Z. mays*, *D. carota*, *Arabidopsis*, *D. carota*, pak choi and *N. officinale*. In the phylogenetic tree MYB60 proteins of different species are divided into two larger groups *SlMYB306* and *ZmMYB306*. The shared position in a single branch indicates that the two genes are genetically close though they belong to different species (Figure 2). Furthermore, the results of evolutionary analysis show that *AtMYB60* of *Arabidopsis*, *NoMYB60* of *N. officinale*, *BcMYB60* of pak choi, *SlMYB306-like* of *S. lycopersicum* and *DcMYB306* and *DcMYB306-like* of *D. carota* are closely related. They are relatively conserved in the evolution process and may originate from the same ancestral gene. In addition, compared with other species, *NoMYB60* of *N. officinale* is more closely related to *AtMYB60* of *Arabidopsis*.

### 3.3. Expression Pattern Analysis of NoMYB60

MYB transcription factors are widely involved in plant secondary metabolism. In order to explore the role of the *NoMYB60* gene in flavonoid synthesis of watercress, samples from different watercress tissues were taken to analyze the expression pattern of *NoMYB60* gene, as shown in Figure 3A. The expression of *NoMYB60* was different in different tissues, and the expression of *NoMYB60* was higher in the petiole, the leaf and especially the root. Furthermore, by analyzing the promoter of the *NoMYB60* gene through the PlantCARE website, we found that the promoter sequence of *NoMYB60* includes one cis-acting element (ABRE) in response to ABA, one cis-acting element (CGTCA) in response to MeJA and two cis-acting elements (TCA elements) in response to SA. Previous studies have shown that ABRE cis-acting elements play an important role in mediating the regulation of ABA [13,14], CGTCA cis-acting elements play an important role in mediating the regulation of MeJA [15] and TCA cis-acting elements play an important role in mediating the regulation of SA [16]. Therefore, we conclude that the *NoMYB60* gene may respond to ABA, SA and MeJA. As shown in Figure 3B–D, our study of the expression profile of the *NoMYB60* gene in watercress under ABA, SA and MeJA treatments using qRT-PCR showed that *NoMYB60* has a trend of increasing first and then decreasing after ABA, MeJA and SA treatments. The expression of *NoMYB60* was the highest at 12 h after ABA treatment, the expression of *NoMYB60* was the highest at 36 h after MeJA treatment and the expression of *NoMYB60* was the highest at 2 h after SA treatment. These results are also consistent with the fact that MYB60 is a transcription factor regulated by ABA, SA and MeJA (Figure 3).

### 3.4. Subcellular Localization of NoMYB60 Protein

35S:GFP and 35S:NoMYB60-GFP fusion expression vectors were constructed (Figure 4A), and transformed into tobacco. After 48–72 h of *Agrobacterium* injection, the localization of NoMYB60 protein was observed and photographed with a confocal laser scanning microscope. The results show that 35S:NoMYB60-GFP fusion protein emits green fluorescence in the nucleus, indicating that NoMYB60 is located in the nucleus (Figure 4).

### 3.5. Virus-Induced NoMYB60 Silencing Caused Increased Flavonoid Content in Watercress

To further verify the effect of *NoMYB60* on the flavonoid synthesis of watercress, virus-induced gene silencing (VIGS) was used to obtain loss-of-function plants. The phenotype of mosaic leaves is emblematic of a silencing plant, and this became increasingly apparent in the positive plants of NoMYB60-PTY and PTY after two weeks. The potential loss-of-function plants were collected, and the silencing efficiency of *NoMYB60* was evaluated by qRT-PCR. As shown in Figure 5, the transcript abundance of *NoMYB60* decreased by more than 80% in NoMYB60-silenced plants #1, #3 and #7 in comparison with the PTY plant. The flavonoid content of NoMYB60-PTY and PTY plants was determined, and the flavonoid content after gene silencing was found to have increased to varying degrees (Figure 5C). This result reveals the function of *NoMYB60* in flavonoid synthesis.

### 3.6. Effect of Virus-Induced NoMYB60 Silencing on Key Genes of Flavonoid Synthesis Pathway

In order to determine whether or not the *NoMYB60* gene is involved in the regulation of the flavonoid synthesis pathway in watercress, molecular detection was performed on the expression of key genes in the flavonoid synthesis pathway in *NoMYB60* transgenic watercress (Figure 6). Compared with PTY plant, the transcription levels of *PAL*, *C4H* and *4CL* genes in the earlier stage of the flavonoid synthesis pathway in NoMYB60-PTY plants were down-regulated to varying degrees, and the expression of *CHs*, *F3H*, *DFR*, *ANS* and *UFGT* genes in the late stage of the flavonoid synthesis pathway were up-regulated to varying degrees, indicating that *NoMYB60* regulates flavonoid synthesis by mediating key genes of the flavonoid synthesis pathway.

### 3.7. NoMYB60 Interacts with NoBEH1/2 In Vitro

In apples, MYB60 interacts with BEH [17]. In order to verify whether or not MYB60 and BEH interact in watercress, we cloned homologous genes *NoBEH1* and *NoBEH2*, and we verified the interaction relationship by Y2H. We first performed the transcription activation assay of the *NoMYB60* gene (Figure 7) and found that yeast cells transferred to pGBKT7-NoMYB60 and pGADT7 can grow normally in SD/−T/−L and SD/−T−L−H−A media, indicating that *NoMYB60* has transcriptional activation activity. According to the structural characteristics of the NoMYB60 protein sequence, it was truncated and retained its N-terminal and intermediate regions (1–277 amino acids), then called *NoMYB60-*1. The results showed that the yeast transformant of pGBKT7-NoMYB60-1 plasmid could not grow on the SD/−T−L−H−A medium, indicating that NoMYB60-1 has no self-activation, and the transcriptional activation region of NoMYB60 was at the C-terminal.

In the Y2H assay, we discovered that co-transformation containing the PGADT7-NoBEH1/2 and pGBKT7-NoMYB60-1 plasmids could grow on both SD/−T−L and SD/−T−L−H−A media. This result indicates that NoMYB60 may physically interact with NoBEH1/2 (Figure 8).

### 3.8. Verification of the Interaction between NoMYB60 and NoBEH1/2 by BiFC

The binding relationship between NoMYB60 and potential interacting proteins was further analyzed by BiFC assay. No yellow fluorescence signal was found in tobacco leaf epidermal cells co-transformed with NoMYB60-cYFP and empty vector nYFP, while significant YFP yellow fluorescence signal was found in tobacco leaf epidermal cells co-transformed with NoBEH1/2-nYFP and NoMYB60-cYFP, indicating that NoMYB60 interacts with NoBEH1/2 in the nucleus of tobacco cells (Figure 9).

## 4. Discussion

Flavonoids are important substances for plants to resist free radicals and antioxidants, and they play a crucial role in the nutritional quality of plants. Therefore, analyzing the biosynthesis of flavonoids is of great significance to germplasm optimization and production practice. In this study, we isolated and cloned a flavonoid biosynthesis-related gene from watercress and named it the *NoMYB60* gene due to its high homology with *Arabidopsis AtMYB60*.

Many studies have shown that the *MYB60* gene is an important promoter in the hormone signal transduction pathway [11,18] and plays an important role in various biological processes, such as promoting root growth and regulating stomatal activity [6,19]. This study found that *NoMYB60* was highly expressed in roots (Figure 3A), which may be related to the gene’s involvement in stomatal regulation as it plays a role in promoting root growth and increasing water absorption. The expression of *NoMYB60* in watercress treated with ABA, SA and MeJA first increased and then decreased, which proved that *NoMYB60* could respond to ABA, SA and MeJA treatment.

The synthesis of flavonoids is affected not only by structural genes, but also by regulatory factors, their own growth and development signals, and the external environment. Many researchers have proved that MYB transcription factors, especially R2R3-MYB transcription factors, play an important role in the flavonoid metabolic pathway. In *Eutrema salsugineum*, overexpression of the *EsMYB90* gene can induce the expression of key flavonoid synthesis enzyme genes such as *PAL* and *CHs*, resulting in the accumulation of anthocyanins [20].In grape, overexpression of the *VvMYBPA1* gene can specifically induce the expression of *LAR* and *ANR* [21], resulting in increased accumulation of tannins. In this study, with the *NoMYB60* gene silenced, the transcription levels of *NoPAL*, *NoC4H* and *No4CL* in the early stage of the flavonoid synthesis pathway were downregulated to varying degrees, and the expression levels of *NoCHs*, *NoF3H*, *NoDFR*, *NoANS* and *NoUFGT* genes in the later stage of the flavonoid synthesis pathway were upregulated to varying degrees. A similar situation also appeared in *Vitis davidii*. When the *VdMYB14* gene was overexpressed, the expression of the key genes affecting anthocyanin synthesis, *NtLAR* and *NtANR*, was significantly increased, while the expression of the key gene in the anthocyanin synthesis pathway, *NtUFGT*, was significantly downregulated [22]. Similar to the above-mentioned plant MYB transcription factor, *NoMYB60* has different transcriptional regulation effects on each enzyme gene of the flavonoid biosynthesis pathway.

The *MYB60 gene has different* biological functions in different plants. For example, overexpression of the *AtMYB60* gene in *Lactuca sativa* results in leaf decolorization [23]. However, in *Arabidopsis*, *V. vinifera* and *S. lycopersicum*, and loss-of-function mutation of AtMYB60 causes stomatal closure, thereby increasing resistance to drought, but without any phenotypic effect [6,24,25]. In order to elucidate the biological function of the *NoMYB60* transcription factor in watercress, we first demonstrated that *NoMYB60* is a transcription factor localized in the nucleus by subcellular localization (Figure 4). To further analyze the regulatory role of *NoMYB60* in the flavonoid metabolic pathway, we constructed a NoMYB60-PTY silencing vector using VIGS technology and transformed watercress using the gene gun method (Figure 5). Through RT-qPCR detection (Figure 6), it was found that after the virus-induced silencing of *NoMYB60*, the expression levels of upstream key enzyme genes *NoPAL*, *NoC4H* and *No4CL* in the flavonoid metabolic pathway were downregulated, and the expression levels of the downstream enzyme genes *NoCHs*, *NoF3H*, *NoDFR*, *NoANS* and *NoUFGT* were upregulated, resulting in increased accumulation of flavonoids. Therefore, we speculate that *NoMYB60* negatively regulates flavonoid biosynthesis by inhibiting the expression of key genes such as *NoCHs*. In addition, the transcriptional activation activity assay of *NoMYB60* indicated that *NoMYB60* is a transcriptional activator. Meanwhile, this study verified the interaction between NoMYB60 and NoBEH1/2 by Y2H and BiFC. Studies have shown that the BZR family transcription factor BEH is involved in BR-regulated flavonoid biosynthesis [17]. Therefore, we speculate that NoMYB60 may participate in BR signal transduction by interacting with NoBEH1/2, thereby regulating the biosynthesis of flavonoids in watercress. However, the specific synthesis mechanism of the *NoMYB60* gene in flavonoid synthesis in watercress remains to be further studied.

## 5. Conclusions

The full-length coding sequence of the *NoMYB60* gene was cloned from watercress, and its structure and function were analyzed. *NoMYB60* has a high expression level in the roots, leaves and petioles in response to ABA, SA and MeJA. Virus-induced gene silencing assays demonstrated that *NoMYB60* participates in the biosynthesis of flavonoids. Furthermore, the Y2H and BiFC assays showed that NoMYB60 proteins display transcriptional activation activity and interact with NoBEH1/2. These findings contributed to our further understanding of MYB60 in cruciferous plants and provided a solid foundation for further research on the regulatory mechanism of flavonoid synthesis in *Nasturtium* or other species.

## Figures and Tables

**Figure 1 genes-13-02109-f001:**
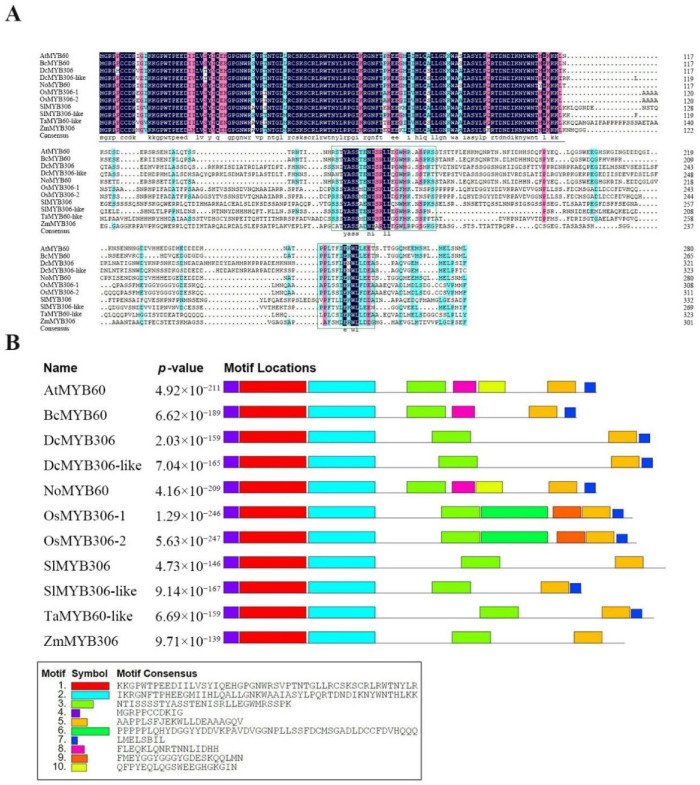
Multiple sequence alignment and structure analysis of MYB60 protein. (**A**) Multiple sequence alignment of MYB60 protein in different species, including *Arabidopsis* (At), pak choi (Bc), *N. officinale* (No), *O. sativa* (Os), *Z. mays* (Zm), *S. lycopersicum* (Sl), *D. carota* (Dc) and *T. aestivum* (Ta). The conserved R2R3 domain sequence is marked above the purple line and the conserved Box1 and Box2 sequences are marked in the green box. (**B**) MYB60 protein has a conservative motif distribution, and the related sequence information of ten motifs is at the bottom of the picture.

**Figure 2 genes-13-02109-f002:**
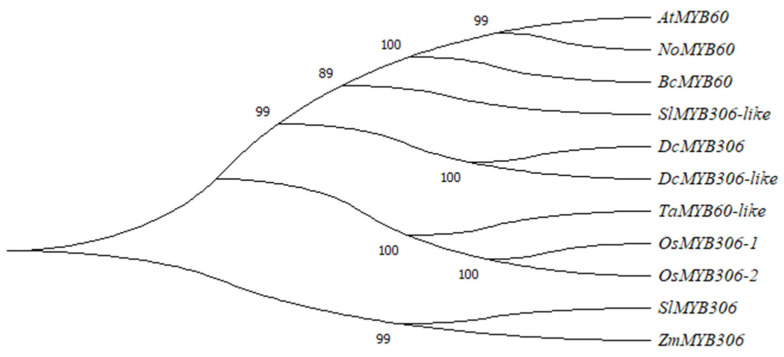
Phylogenetic analysis of *MYB60* genes in different species. The phylogenetic tree of MYB60 was constructed by MEGA7.0, including *Arabidopsis* (At), pak choi (Bc), *N. officinale* (No), *O. sativa* (Os), *Z. mays* (Zm), *S. lycopersicum* (Sl), *D. carota* (Dc) and *T. aestivum* (Ta).

**Figure 3 genes-13-02109-f003:**
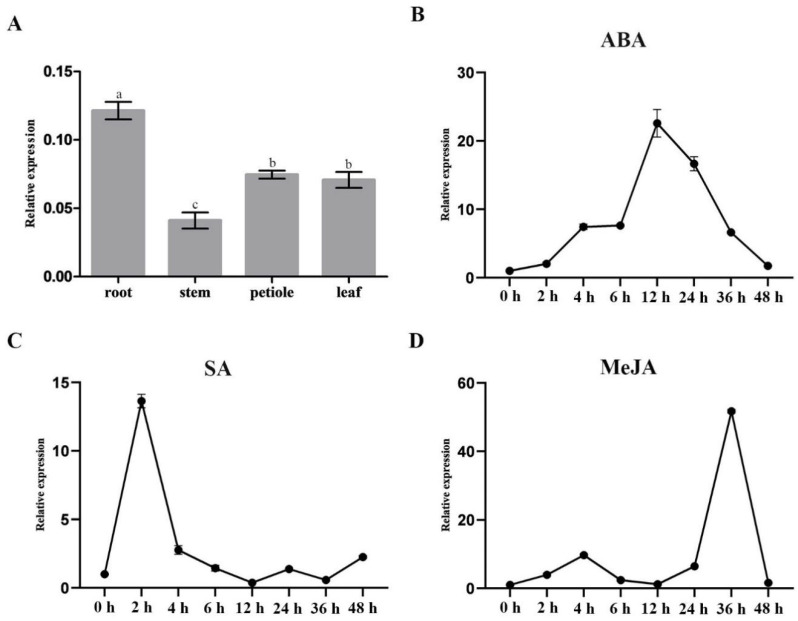
Expression profile of *NoMYB60* gene in watercress. (**A**) *NoMYB60* expression levels in different tissues (root, stem, petiole and leaf). The error lines represent the standard errors, different letters represent significant differences, *p* < 0.05. (**B**–**D**) expression analysis of *NoMYB60* under respect ABA, SA and MeJA treatments. Watercress, cut to a length of about 15 cm, was exposed to ABA (5 mg/L), SA (5 mg/L) and MeJA (1 mg/L) treatments over a consecutive time course (0, 2, 4, 6, 8, 12, 24, 36 and 48 h). The horizontal axis represents processing times and the vertical axis represents the relative expression.

**Figure 4 genes-13-02109-f004:**
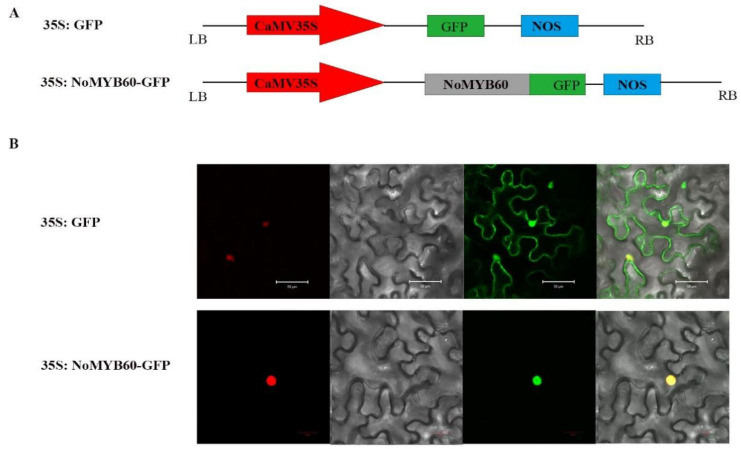
Subcellular localization of NoMYB60. (**A**) The construct of 35S:NoMYB60-GFP fusion protein; (**B**) localization of 35S:NoMYB60-GFP in tobacco leaves was observed under a laser confocal microscope. The panels from left to right show mCherry (nuclear marker), bright field, fluorescence and merged fluorescence images.

**Figure 5 genes-13-02109-f005:**
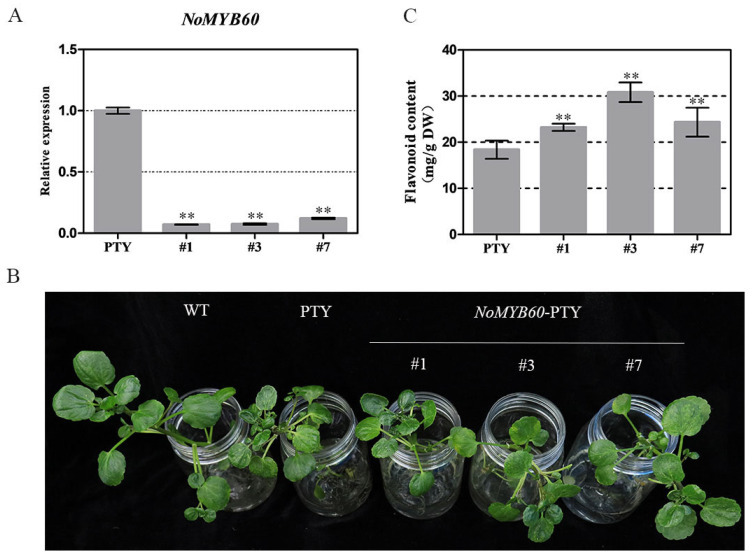
Virus-induced *NoMYB60* silencing in watercress. (**A**) Relative expression of *NoMYB60* in silent plants. Error bars represent standard errors. (**B**) Plant phenotype after silencing of *NoMYB60* gene. (**C**) Flavonoid content in silenced plants (**, *p* < 0.05).

**Figure 6 genes-13-02109-f006:**
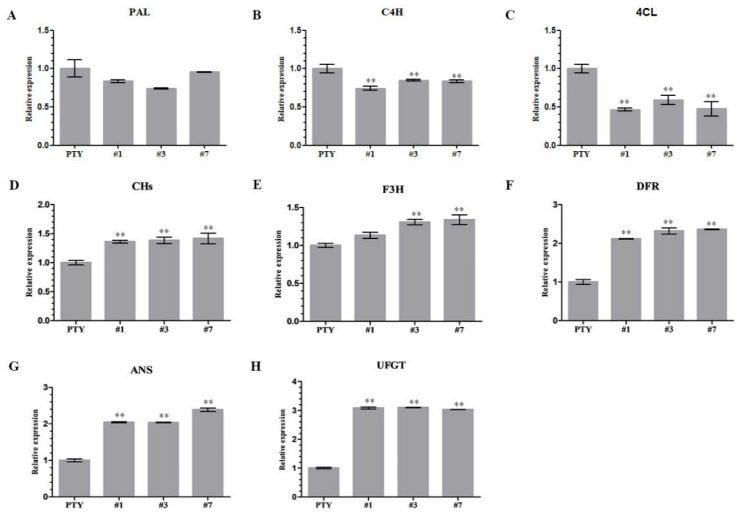
Effects of *NoMYB60* silencing in watercress on transcription levels of genes related to flavonoid metabolism in watercress. (**A**) PAL: phenylalanine ammonia-lyase; (**B**) C4H: cinnamic acid 4-hydroxylase; (**C**) 4CL: 4-coumarate:CoA ligase; (**D**) CHs: Chalcone synthase; (**E**) F3H: flavanone 3-hydroxylase; (**F**) DFR: dihydroflavonol-4-reductase; (**G**) ANS: anthocyanin synthase; (**H**) UFGT: UDP-glucose:flavonoid 3- O-glucosyltransferase; ** represent *p* < 0.05.

**Figure 7 genes-13-02109-f007:**
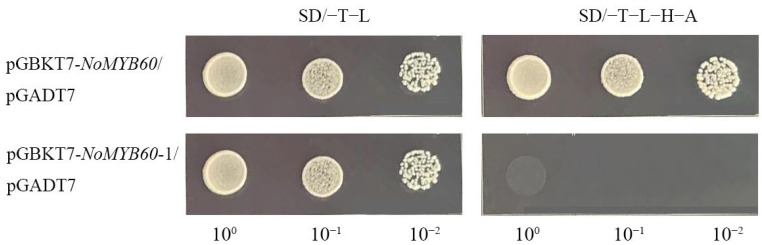
Transcriptional activation activity analysis of NoMYB60. Transcription activity assay of NoMYB60 in yeast. NoMYB60 and NoMYB60-1 represent the full-length protein, as well as the regions encoding 1–277 amino acids of NoMYB60, respectively, which were inserted into the pGBKT7 vector.

**Figure 8 genes-13-02109-f008:**
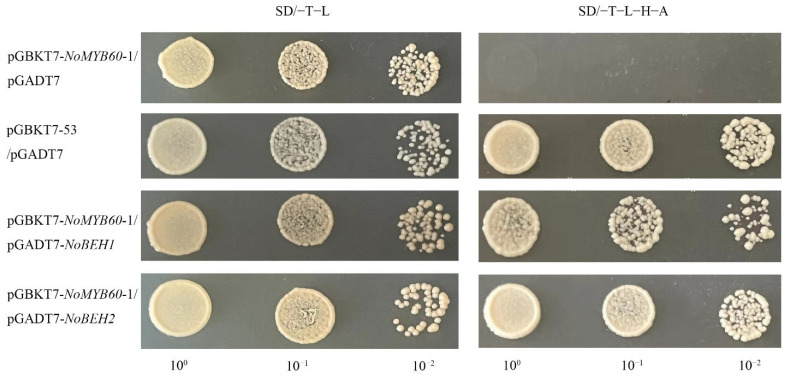
Yeast two-hybrid analysis of interaction between NoMYB60 and NoBEH1/2. The verification of the interaction between NoMYB60-1 and NoBEH1/2 by Y2H assay. The yeast cells were grown on the SD media: SD/−T−L (left), SD/−T−L−H−A (right). pGBKT7-53/pGADT7 were used as the positive control. 10^0^, 10^−1^ and 10^−2^ represent the concentrations of yeast solutions.

**Figure 9 genes-13-02109-f009:**
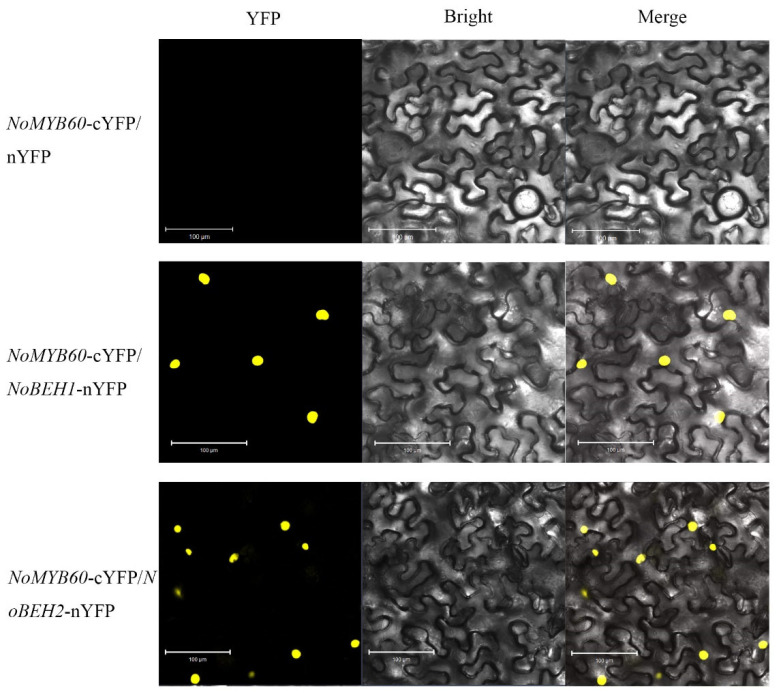
BiFC analysis of the interaction between NoMYB60 and NoBEH1/2 in tobacco leaves. The yellow fluorescence signal is displayed in a dark field, bright field and merged, respectively. Scale bar, 100 µm.

**Table 1 genes-13-02109-t001:** Information on *NoMYB60* promoters.

Promoter Length	Element Name	Number of Elements	Position of the GARE Element
2000 bp	ABRE	1	“−” Strand: 391–395
	CGTCA-motif	1	“+” Strand: 762–766
	TCA-element	2	“−” Strand: 403–407 “−” Strand: 604–608

## Data Availability

Data are available on request.

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
