# Peer review of "Cloning and Functional Analysis of NoMYB60 Gene Involved in Flavonoid Biosynthesis in Watercress (Nasturtium officinale R. Br.)"

_genes, 2022, doi:10.3390/genes13112109_

Round 1
Reviewer 1 Report
This study identified and characterized NoMYB60 that negatively regulates flavonoid Biosynthesis in watercress. Overall, they have provided some useful data but are incomplete nor sufficient. The authors need to address the following questions and concerns.
Major comments:
- You may need to give some introduction about flavonoids.
- Please describe what flavonoids are accumulated in watercress and quantify each flavonoid accumulation in your VIGS plants and WT by LCMS. This is because the key conclusion of your manuscript (NoMYB60 negatively regulates flavonoid Biosynthesis in watercress) relies on the VIGS data, but they are completely not well performed. The authors also did not describe how they measure flavonoid content.
- For gene expression analysis, a lot of key genes (eg CHI, F3’H), are missing. First of all, please show a flavonoid biosynthetic pathways indicating what flavonoids are accumulated in watercress and indicate the genes required for their biosynthesis. Then, please carry out gene expression analyses for all of the genes.
- What are the target genes that NoMYB60 that regulate? It would be good to test whether NoMYB60 can bind to the promoters of those flavonoid biosynthetic genes you tested.
- Line 337-341: This part is opposite to your results. Please revise.
Other comments:
- Please check the whole manuscript for mistakes (not language issue. It’s about the content). There are just too many.
- Please complete the figure legend and include all the required information. Too many essential information for understanding standing the figures are missing. As there are too many to name, please check and complete by yourself.
Reviewer 2 Report
The paper of Ma et al. deals with the cloning of gene coding for MYB60, a transcription factor negatively regulating flavonoid biosynthesis. Although many experiments were performed, the paper is not sufficiently well written to be reviewed in its present form and should be revised entirely in a more correct and fluent form.
Here are some advises but the reviewer cannot correct all.
The abstract has to be entirely revised as it contains many mistakes such as “MYB60 gene belongs to the R2R3-MYB subfamily and contains MYB31/30/96/94 genes”, “MYB60 transcription factor was isolated from watercress”, ” In conclusion, NoMYB60 negatively regulates the synthesis of flavonoids by affecting the expression of genes”
Remove the “for the first time”
The introduction is much too short, quite vague and with mistakes
The material and method is sometimes presented as protocols and not complete (ex flavonoid testing?, Y2H, ..., cloning by homologous recombination)
Genes abbreviated, should be full name the first time cited.
Results
Line 145 On what basis it is an unstable protein?
Line 152: how can you calculate % of homology (similarity? Identity?) based on the multiple alignment
Figure 1: what are these motifs? Are they functional motifs? The yellow one is only present in At and No. The alignment is not readable
Figure 2: As At and No are both Brassicaceae, it is normal that MYB60 proteins are more closely related than the other species.
Line 194: higher than what?
Line 196: size of the promoter sequence analyzed?
Line 215: soybean flaps???
3.4: protocol
Discussion has to be improved
Conclusion is like the abstract...
Etc etc...
We are in 2022 and not 2021...
Reviewer 3 Report
The manuscript entitled as “Cloning and functional analysis of NoMYB60 gene involved in flavonoid biosynthesis in watercress (Nasturtium officinale)” describes cloning and functional analysis of NoMYB60, a MYB transcription factor gene from watercress. The authors investigated the role of NoMYB60 in the context of regulation of flavonoid biosynthesis. Different attributes of NoMYB60 were studied including sequence structure analysis, expression analysis in response to different hormones such as ABA, SA, and MeJA, and its subcellular localization. Further the functional role of NoMYB60 was investigated using gene silencing (VIGS). Taken together, it was concluded that NoMYB60 transcription factor negatively regulates the synthesis of flavonoids by affecting the expression of genes. Altogether, the present study provides a valuable knowledge resource for further research that will allow for better understanding of the regulatory mechanism of flavonoid biosynthesis.
However, my specific commnets are mentioned in the below points:
1. To analyze the evolutionary relationship of MYB60 protein among diverse plant species, a phylogenetic tree at the amino acid level was constructed with MEGA 7.0 software.However, the authors have not mentioned anywhere whether they have used maximum likelihood (ML) method or neighbor joining (NJ) method.
2. In this study the authors have carried out the expression profile of NoMYB60 in response to differentplant hormone treatments (ABA, SA, and MeJA). On what basis the concentration of plant hormones, age of plants for harvesting, and time-points of sampling were chosen?
3. The expression of NoMYB60 was found to be the highest at 12h and 2h after ABA and SA treatment respectively; however, after that the expression of NoMYB60 decreases. How this trend of the expression can be explained? (OnFigure3)
4. The qRT-PCR analysis showed that the transcript levels of PAL, C4H and 4CL genes in NoMYB60-PTY silenced plants were down regulated to varying degrees whereas, the expression level of CHS, F3H, DFR, ANS and UFGT genes in the late stage of flavonoids synthesis pathway were up regulated to varying degrees, reflecting that NoMYB60 regulates flavonoids synthesis by mediating thekey genes of flavonoid biosynthesis pathway. Furthermore, the total flavonoid content in silenced plants was also found to be increased as compared to the respective control. What was the impact of the gene silencing on specific flavonoids, e.g. anthocyanin?
5. The author has carried out VIGS for gene silencing with only vector control but the authors should have also taken an endogenous control either PDS or CHS which provides a visual indication of the success of silencing and, as such, allows the calculation of the percentage silencing frequency, effectiveness, and efficiency.
Round 2
Reviewer 2 Report
The manuscript has been improved.
Several minor changes have to be made and the figure 1 should be of better quality.
Line 23 The subcellular localization of NoMYB60-GFP?
Line 34 Transactivation
Line 44 reference for the number of flavonoids
Line 46: Phenylalanine ammonia-lyase
Line 50: reference what for etc???
Line 55: gene name missing
Line 60 : reference
Line 70 reference number
Line 108 : chinese in the table!
Figure 1 still of bad quality, should be improved
Line 246: what plant material for the treatment?
Legend line 257: written as protocol
Line 279 silenced plants
Line 286 police
Line 297 and paragraph 3.6: plant material?? Age??
